# The Characteristics of Badminton-Related Pain in Pre-Adolescent and Adolescent Badminton Players

**DOI:** 10.3390/children10091501

**Published:** 2023-09-02

**Authors:** Xiao Zhou, Kazuhiro Imai, Zhuo Chen, Xiaoxuan Liu, Eiji Watanabe, Hongtao Zeng

**Affiliations:** 1School of Physical Education, Huazhong University of Science and Technology, Wuhan 430074, China; syuu@hust.edu.cn (X.Z.); zenghongtao@hust.edu.cn (H.Z.); 2Department of Life Sciences, Graduate School of Arts and Sciences, The University of Tokyo, Komaba, Meguro-ku, Tokyo 1538902, Japan; chenzhuo11111@gmail.com; 3Faculty of Medicine, Dalhousie University, Halifax, NS B3H 4R2, Canada; luluxxliu@gmail.com; 4Institute of Sport, Senshu University, Kawasaki 2148580, Japan; watana@isc.senshu-u.ac.jp

**Keywords:** sports-related pain, pain incidence, racquet sports, child

## Abstract

Body pain, often considered as an early sign of injury in young players, warrants thorough study. This study aimed to examine the distribution of badminton-related pain and prevalence in pre-adolescent and adolescent badminton players. Profiles of badminton-related pain were surveyed using a questionnaire among 366 pre-adolescent and adolescent badminton players aged 7–12 years. The distribution of badminton-related pain was described, and the pain incidence was calculated. Proportions of pain per 1000-training-hour exposures were the main outcome measures. The analysis considered various age groups (7–8, 9–10, and 11–12 years) and years of badminton experience (≤2, 2–3, and > 3 years). In total, 554 cases of badminton-related pain were reported. The ankle was the most common site, followed by knee, plantar, shoulder, and lower back. The overall pain rate per 1000-training-hour exposure was 3.06. The 11–12-year-old group showed the highest pain rate, significantly greater than the 7–8-year-old group and the 9–10-year-old group. Additionally, the prevalence of pain exhibited an increasing trend with age. Finally, regardless of the age groups, participants with 2–3 years of badminton experience had the highest pain rate. These findings might help inform targeted interventions to reduce the high prevalence of pain in various body regions across pre-adolescent and adolescent badminton players.

## 1. Introduction 

Sports injury prevention is highly recommended for child players as childhood sports have dramatically changed in recent decades. There is now a strong emphasis on early sport specialization and achieving competitive success, which has led to increased pressure to commence high-intensity training at a young age. This early training can potentially contribute to gradual-onset injuries and career burnout [1,2]. Badminton requires repetitive forehand overhead motion, lunges, jumps, quick directional changes, and the use of major body joints in response to different types of strokes [3,4]. Similar to other overhead sports, the forehand overhead motion is a crucial skill during badminton training and matches. Previous studies reported that during the performance of an overhead motion, the trunk transfers 50% of the total energy generated by the ground reaction from the lower limbs (including ankle and knee) to the upper limbs (including shoulder, elbow, and wrist), whereas 13% of the work is completed by the shoulder [5]. To produce energy, transfer body weight, and maintain balance, the knee plays a pivotal role during badminton play. Due to the specific characteristics of badminton, pain and injuries involving various body joints, especially the knee, lower back, and shoulder, can occur [6,7,8]. Epidemiological studies of badminton have reported that the prevalence of sustaining at least one badminton injury varies from 49.6% to 82% in youth badminton players [6,7,8]. In recent years, the injury rate per 1000 athlete-hours of exposure has been used to study sports-related injuries in numerous studies [6,9,10]. These studies have reported that the incidence rate of badminton injuries per 1000 badminton-hour exposures varies from 1.55 to 1.82 injuries in 7–18-year-old badminton players [6,9].

Body pain might be an early sign of a potential high-risk bone stress injury, so attention should be paid to athletes reporting pain in various body regions [2,11]. Literature reports have indicated that in badminton players aged 6–18 years, shoulder pain was reported by 27.6% of all badminton players and lower back pain was reported by 35.4% of all badminton players. Pain related to badminton was common in the foot, knee, shoulder/elbow, and back [12,13]. Another previous study on badminton injury and pain among Japanese elite university badminton players showed that pain related to badminton mostly occurred in the shoulder, followed by the lower back, foot, thigh, knee, elbow, and ankle [14]. Moreover, pain has other negative influence on badminton players as well. For instance, past studies revealed that shoulder pain caused by instability, scapulothoracic dyskinesia, or subacromial impingement [15] would negatively affect performance during badminton playing and influence activities of daily living, such as by disturbing sleep [16]. In addition, more than one-third of the badminton players who complained of shoulder pain continued to practice or play, which might increase the likelihood of sustaining a shoulder injury [16]. Meanwhile, in badminton players, a significant association existed between shoulder pain, lower back pain, and knee pain [13]. This suggests that pain localized in one of the three anatomic sites will produce a load that has to be compensated for by the movement of the other two sites. Nonetheless, studies on the epidemiology of pain related to badminton in the knee, lower back, and shoulder among pre-adolescent and adolescent badminton players are limited. Furthermore, studies on pain related to badminton per 1000-training-hour exposure have not been found [4].

Therefore, enhancing our understanding of pain is crucial for injury prevention and long-term health. However, the literature that has explored the epidemiology of pain in youth badminton players is limited. This study aimed to survey the distribution and rate of badminton-related pain, particularly focusing on knee pain, lower back pain, and shoulder pain, among pre-adolescent and adolescent badminton players participating at the national tournament level.

## 2. Materials and Methods

A cross-sectional study was designed to investigate pre-adolescent and adolescent badminton players who ranged in age from 7 to 12 years. All the participants were members of the Japan Schoolchildren Badminton Federation. We recruited 611 pre-adolescent and adolescent badminton players (260 boys and 351 girls) attending the national tournament games at three randomly selected locations in 2019. All participants had the consent of their guardians. They, along with their guardians, completed a custom-designed questionnaire survey that was modified from previous studies [17,18] during the period of the national tournament games. The custom-designed questionnaire consisted of two parts (See Appendix A). In the first part, information was collected regarding sex, age, dominant side, years of badminton-playing experience, badminton training duration per day, warm-up and cool-down practices, and the number of days dedicated to badminton training weekly. We defined the time spent on badminton skill training or somatic training under the coach’s supervision as training hours. The duration of the warm-up and cool-down periods was not considered training-exposure time. In the second part, pain and injuries related to badminton over the past 12 months were surveyed. All the pain and injuries were reported specifically in terms of 25 anatomical regions presented in a body image (including face, chest, abdomen, shoulder, elbow, wrist, finger, groin, quadriceps, knee, ankle, shin, toe, head, neck, scapular, back, upper arm, forearm, lower back, hip, hamstring, calf, Achilles tendon, and plantar) (See Appendix A), pain/injury type (pain, acute injury, gradual-onset injury), and the age at which they occurred.

In this study, we established the judgment criteria for sports injury/pain in accordance with the International Olympic Committee to maintain uniformity in definitions and allow data from other studies to be compared [19]. Pain was defined as any painful physical discomfort (ache or soreness in anatomical regions, without or with radiating pain) with sustained sports capability [20] as follows: (1) being able to continue the present badminton training session or match; (2) participating in the next scheduled badminton training session or match without any time loss; (3) not needing medical care during and after the badminton training session or match. An injury was defined as any physical discomfort that caused one or more of the following three judgment criteria to be met during training or match play: (1) having to immediately stop the present badminton training session or match; (2) being absent from the following badminton training session or match; and/or (3) the need for medical attention regardless of the possibility of missing a training session or a match. Participants who reported injured or surgically treated sites without any pain sites were excluded. Injured or surgically treated sites were excluded once injured or surgery sites and other pain sites were both reported. Additionally, participants with less than one year badminton experience were excluded. For data analysis, we categorized all the participants into three groups based on their ages: the 7–8-year-old group, the 9–10-year-old group, and the 11–12-year-old group.

The Graduate School of Arts and Sciences, the University of Tokyo, Japan reviewed and approved this study (Notification Number 602-2 26 July 2018) in compliance with the declaration of Helsinki statement.

### Statistical Analysis

We used medians with an interquartile range (IQR) to analyze the distribution of the participants based on badminton experiences. The three groups were further divided into subgroups, categorized according to the distribution of badminton experiences.

The normality of baseline parameters was examined by the Shapiro–Wilk test. The data for the baseline parameters, including age, years of badminton-playing experiences, daily badminton-training hours, weekly badminton-training days, weekly badminton-training hours, and yearly badminton-training hours, exhibited non-normal distribution. A Kruskal–Wallis ANOVA followed by Dunn’s test was applied for non-parameter statistical analysis of the three groups. Poisson distribution was used to calculate the pain rate per 1000-training-hour exposure for comparing the pain incidences between the three groups. The 95% confidence interval (CI) of the pain rate per 1000-training-hour exposure was also calculated. An hour of badminton-training exposure is defined as one hour of somatic condition training or badminton skills training under the supervision of the coach without warm-up and cool-down time by a single badminton player. We calculated the pain rate per 1000-training-hour exposure in the badminton training period as follows:Pain rate per 1000-training-hour exposure = [∑(No. of pains)/∑{(No. of participants) × (hours of badminton training)}] × 1000.

If the 95% CI did not overlap, significant differences in pain rate per 1000-training-hour exposure were assumed to exist between the groups. Statistical significance was defined as *p* < 0.05.

## 3. Results

Overall, 366 out of 611 participants aged 7–12 years, including 164 boys and 202 girls, met the inclusion criteria. The average age of the 366 participants was 10 years, with a standard deviation of 1.25 years. Sixty-nine percent (223 participants: 87 boys and 136 girls) of all the 366 participants experienced at least one badminton-related pain. Based on the age distribution, all the participants were divided into three groups: the 7–8-year-old group (*n* = 52), the 9–10-year-old group (*n* = 155), and the 11–12-year-old group (*n* = 159). The baseline parameters of the three groups are presented in Table 1. There were 28 male participants and 24 female participants in the 7–8-year-old group (right hand: 48 participants, left: 4 participants), 69 male participants and 86 female participants in the 9–10-year-old group (right hand: 131 participants, left: 24 participants), and 67 male participants and 92 female participants in the 11–12-year-old group (right hand: 143 participants, left: 16 participants). Among the three groups, significant differences in age (*p* < 0.001) and years of badminton-playing experience (*p* < 0.001) were observed, whereas no significant differences in other variables, including training hours per day, training days per week, training hours per week, and training hours per year, were observed. Additionally, with respect to warm-up and cool-down routines, all the participants in the 7–8-year-old group performed a warm up, while 94.8% of the participants in the 9–10-year-old group and 93.1% of the participants in the 11–12-year-old group did so. The participants in the three groups who performed a cool down were slightly fewer than those who performed a warm up: 75% in the 7–8-year-old group, 68.4% in the 9–10-year-old group, and 67.9% in the 11–12-year-old group.

In total, 554 cases of pain related to badminton were reported at the 25 survey anatomic sites. The overall pain rate per 1000-training-hour exposure was 3.06 (95% CI: 2.81–3.32). All the pain incidences in the face, chest, abdomen, head, neck, scapular, back, forearm, and hip constituted less than 2% of the total; therefore, these nine anatomic sites were grouped together in the category “others”. As shown in Figure 1, the ankle was the most common pain site, accounting for 12.5% of all reported pain, followed by the knee (11.4%), plantar (10.1%), shoulder (8.1%), lower back (6.5%), toe (5.8%), groin (4.9%), quadriceps (4.9%), elbow (4.5%), calf (4.3%), Achilles tendon (4.3%), hamstring (4.0%), wrist (3.4%), shin (2.7%), and upper arm (2.2%). To improve the understanding of pain related to badminton occurring in the knee, lower back, and shoulder, the pain rates per 1000-training-hour exposure of the three anatomic sites were also calculated. These rates are shown in Figure 2. The overall pain rate of the three sites was 0.80 (95% CI: 0.67–0.93) pain per 1000-training-hour exposure. More specifically, the pain rates for shoulder pain, lower back pain, and knee pain were 0.25 (95% CI: 0.18–0.32) pain, 0.20 (95% CI: 0.13–0.26) pain, and 0.35 (95% CI: 0.26–0.43) pain per 1000-training-hour exposure.

For further analysis, we divided the three groups into three subgroups (≤2, 2–3, and >3 years groups) based on badminton experience distribution using medians with IQR. As shown in Table 2, the pain rate per 1000 h of badminton training significantly increased with age (7–8-year-old group vs. 9–10-year-old group vs. 11–12-year-old group: 1.22 vs. 2.29 vs. 4.34). Regardless of the three groups, participants with 2–3 years of badminton-playing experience presented the highest pain rate per 1000-training-hour exposure.

## 4. Discussion

Pain management is recommended for skeletally immature child athletes to ensure long-term success in sports [2,6]. Considering the tendency towards early badminton specialization, multifaceted training, and frequent competition, this study focused on body pain related to badminton as the primary outcome of interest. The results showed that 60.9% of all the 366 pre-adolescent and adolescent badminton players aged 7–12 years experienced at least one pain associated with badminton, which is less than the pain rate (73%) observed in competitive and amateur badminton players over 18 years [21]. This study also showed that the rate of pain significantly increased with increasing age among 7–12-year-old badminton players.

Previous studies focusing on elite university badminton players aged 18–22 years revealed that badminton-related pain frequently localized in the shoulder (13.7%), lower back (13.2%), foot (11.7%), thigh (10.8%), knee (9.7%), elbow (9.2%), and ankle (9.0%) [14]. However, a past study on the overhead motion sports found different pain patterns among 6–15-year-old badminton players. In that study, the foot was the most frequent pain site, followed by the knee, while the shoulder/elbow ranked the third, and the back ranked the fourth [13]. In this study, the results regarding body pain distribution indicated that the most frequent pain site was the ankle, accounting for 12.5% of all reported pain. The knee came the second at 11.4%, followed by the plantar (10.1%), shoulder (8.1%), and lower back (6.5%). Notably, these findings align more closely with the patterns observed in the study of 6–15-year-old badminton players. The difference in anatomical pain sites between our study and the previous study of elite university badminton players may be attributed to age-related factors.

In the realm of youth overhead motion sports, much attention has been given to addressing pain with a particular focus on the shoulder, lower back, and knee injury prevention. For instance, in a study of 7–12-year-old Japanese baseball players, 15.9% of all the players reported shoulder pain [22]. In another large study comprising 7894 Japanese baseball players with an average age of 8.9 years, 8% of them experienced shoulder pain [23]. These findings contradict earlier studies of baseball players, where 35% of players reported shoulder pain, exceeding 6.04% of pitching-related pain in 9–12-year-old American baseball players [11]. Moreover, a previous study of basketball reported that the prevalence of shoulder pain and lower back pain was 4.6% and 12.9%, respectively, in youth basketball players [24]. For baseball players aged 6–15-year-old, the prevalence of lower back pain and knee pain was 8.4% and 13.1%, respectively [25]. In other overhead motion sports, the prevalence of knee pain varied, with rates of14.3% in softball players, 28.6% in handball players, 10.9% in tennis players, and 12.1% in volleyball players [13].

In this study, among all the badminton players, the prevalence of shoulder pain was 12.3%, lower back pain was 9.8%, and knee pain was 17.2%. This study is the first to study badminton-related pain in the shoulder, lower back, and knee using a rate of pain per 1000 training hours in pre-adolescent and adolescent badminton players. The pain rate per 1000-training-hour exposure was 0.25 for shoulder pain, 0.20 for lower back pain, and 0.35 for knee pain. Furthermore, it was observed that ankle pain had the highest prevalence among pre-adolescent and adolescent badminton players in this study.

Badminton, much like other overhead motion sports such as tennis and baseball, necessitates rapid changes in player positions and dynamic movements such as lunges, jumps, twists, and swings. Due to the characteristics of badminton, it is essential to coordinate all body joint movements frequently. These movements include weight shifting, upper limb rotation, trunk rotation, and instant starting and braking in response to receiving a shuttlecock from a variety of directions [3,4]. These findings could explain the prevalence of badminton-related pain localized to most body sites in this study. Moreover, badminton players with less developed techniques tend to present more internal joint rotation in the horizontal plane as well as more inversion joint moment in the frontal plane [26]. Additionally, the participants in this study were pre-adolescent and adolescent badminton players whose physical fitness (including muscle, bone, and neuro fitness) was still developing. This means they might have been unable to respond to the intensive somatic demands of repetitive weight transfer through turning, pivoting, and landing. These actions place extra stress on the ankle joint during badminton training. In addition, vigorous training can lead to fatigue, which in turn increases the risk of ankle sprain in badminton playing [27]. These findings could interpret why the ankle was the most common pain site among pre-adolescent and adolescent badminton players in this study.

In respect to the pain rate of badminton players, no previous studies using the pain rate per 1000-badminton-training-hour exposure have been found. However, the literature using an injury rate per 1000-athlete-hour exposure has indicated that injury incidence rates tend to increase with increasing age in pre-adolescent and adolescent athletes aged 7–12 years [28]. Our study made a similar observation regarding the pain rate per 1000-training-hour exposure, which also increased with age. Specifically, we observed pain rates of 1.22 in 7–8-year-old badminton players, 2.29 in 9–10-year-old badminton players, and 4.34 in 11–12-year-old badminton players. Another interesting finding was that, regardless of age groups, pre-adolescent and adolescent badminton players with 2–3 years of badminton experience exhibit the highest pain rate. Pre-adolescent and adolescent badminton players typically spend a year learning and practicing badminton skills before progressing from the initial stage to the first supervised training. Then, it takes another year to advance from supervised training to participating in their first competition [29]. We speculated that this transition to competition might lead to high-intensity and aggressive training, even though pre-adolescent and adolescent badminton players may still possess immature badminton skills. This, in turn, could contribute to the increased prevalence of pain in the subsequent year. For future research, it may be valuable to investigate the mastery of badminton skill and training loading among pre-adolescent and adolescent badminton players with 2–3 years of badminton experience.

The findings of this study emphasize that badminton-related pain is a frequent occurrence among badminton players aged 7–12 years. Pain was observed in various anatomical sites, with the most common being the ankle, knee, shoulder, and lower back. To prevent injury and ensure pain/injury-free badminton participation, a prevention approach should be implemented from the pre-adolescent stage. Previous studies have revealed that there are significant differences between extra hamstring muscle flexibility and badminton-related pain, as well as weak static balance ability and badminton-related pain [14]. Given the musculoskeletal and nervous system maturation levels of pre-adolescent and adolescent badminton players, it is essential to develop hamstring muscle flexibility and static balance ability. These efforts may help lessen the prevalence of pain related to badminton. In terms of hamstring muscle flexibility training, Nordic hamstring exercises have been revealed to be effective in boosting hamstring muscle tightness in other overhead motion sports [30]. With respect to static ability training, plyometric training on a balance could improve the static balance ability [31]. Such training may contribute to a decrease in the prevalence of ankle and knee pain. In addition, it is important to acknowledge that improper motion skills can serve as mechanisms for sports injuries and pain [32,33,34]. Coaches should place more emphasis on motor-skill learning among pre-adolescent and adolescent badminton players.

This study has some limitations. Firstly, despite our efforts to minimize potential biases, this study’s retrospective design may introduce recall bias, as participants may not accurately recall all instances of badminton-related pain. Secondly, we did not investigate the specific mechanism leading to badminton-related pain, such as types of movements that triggered the pain. Thirdly, although pain related to badminton has been studied, we failed to find a relationship between pain and a certain injury in the pre-adolescent and adolescent badminton players. A prospective study should be performed to identify whether pain is a risk factor for badminton-related injury in future. Fourth, a follow-up study in baseball demonstrated that pitching counts contribute to shoulder and elbow pain among pre-adolescent and adolescent baseball players [11]. The American Sports Medicine Institute has even designed pitching guidelines including rest recommendations and pitching count limits according to decades of research for baseball players [35]. In contrast, there were no such training guidelines for pre-adolescent and adolescent badminton players. Given that our study found that participants with 2–3 years of experience were more vulnerable to sustaining pain related to badminton, there may be a need for similar guidelines in the context of badminton. Future research should investigate appropriate training volume for badminton-related pain/injury prevention in youth badminton players. Additionally, it would be valuable to assess the severity of pain related to badminton, as it could help us understand the characteristics of badminton-related pain and injuries. Finally, intrinsic factors, such as overhead motor skills and physical fitness, have been identified as contributing factors to pain and injuries in overhead motion sports. For example, in badminton, university players with extra hamstring flexibility showed a higher incidence of badminton injuries [20]. In baseball, pitchers who deliver the ball with a greater arm slot angle may be at increased risk of shoulder injury [32]. Therefore, the physical fitness and badminton motor skills of pre-adolescent and adolescent badminton players should be the focus of future research.

## 5. Conclusions

In pre-adolescent and adolescent badminton players aged 7–12 years, our study revealed that badminton-related pain predominately affected the lower limbs. Furthermore, the prevalence of pain increased with age. These findings might help target the high prevalence of pain in various body regions among pre-adolescent and adolescent badminton players.

## Figures and Tables

**Figure 1 children-10-01501-f001:**
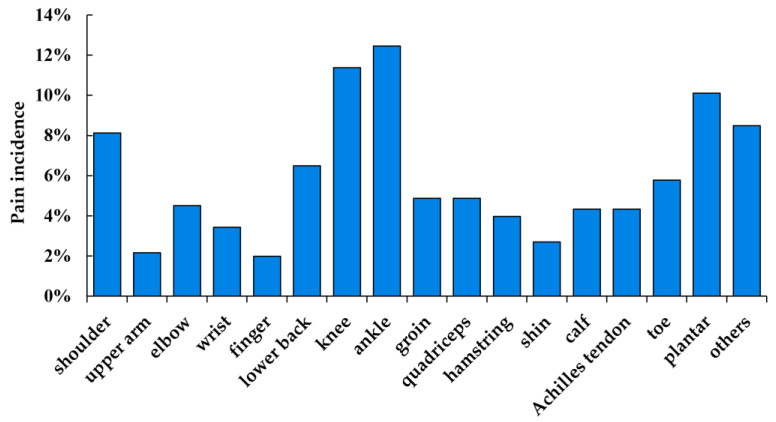
The distribution of pain in all the participants.

**Figure 2 children-10-01501-f002:**
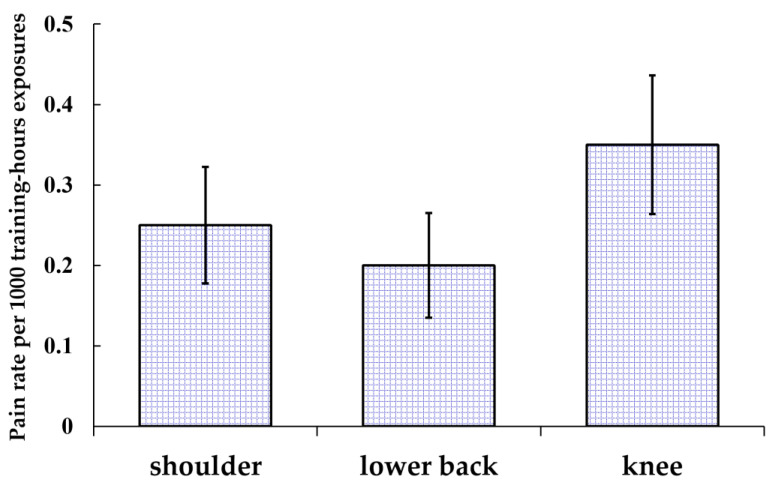
Pain rate per 1000-training-hour exposure in shoulder, lower back, and knee. The error bars represent 95% CI.

**Table 1 children-10-01501-t001:** Baseline parameters of all the badminton players in the three age groups.

Variables	7/8-year-old (*n* = 52)	9/10-year-old (*n* = 155)	11/12-year-old (*n* = 159)
Gender, male/female	28/24	69/86	67/92
Age, y	7.73 ± 0.45 *	9.59 ± 0.49 *	11.15 ± 0.36 *
Dominant side, right/left	48/4	131/24	143/16
Experience, y	2.02 ± 0.86 *	2.73 ± 1.22 *	3.18 ± 1.38 *
Training, hours	2.67 ± 1.12	2.82 ± 0.72	2.74 ± 0.94
Days, per week	4.52 ± 1.40	4.32 ± 1.25	4.57 ± 1.10
Hours, per week	11.86 ± 4.86	12.16 ± 4.75	12.72 ± 6.02
Hours, one year	474.42 ± 194.56	486.58 ± 190.06	508.93 ± 240.65
Warm-up, yes/no	52/0	147/8	148/11
Cool down, yes/no	39/13	106/49	108/51

Values are mean ± standard deviation. * *p* value < 0.001 among the three age groups.

**Table 2 children-10-01501-t002:** Pain rates in different age groups broken down using badminton experience.

Variable	Players	Pain	Training-Hours of Exposures	Pain Rate per 1000 Training-Hour Exposures	95% CI
7–8-year-old					
Overall	52	30	24,670	1.22 *	[0.78–1.65]
≤2 years	37	17	16,490	1.03	[0.54–1.52]
2–3 years	11	10	6220	1.61	[0.61–2.60]
>3 years	4	3	1960	1.53	[0.00–3.26]
9–10-year-old					
Overall	155	173	75,420	2.29 *	[1.95–2.64]
≤2 years	55	51	23,990	2.13	[1.54–2.71]
2–3 years	49	68	23,280	2.92	[2.23–3.62]
>3 years	51	54	28,150	1.92	[1.41–2.43]
11–12-year-old					
Overall	159	351	80,920	4.34 *	[3.88–4.79]
≤2 years	37	75	17,100	4.39	[3.39–5.38]
2–3 years	54	156	26,030	5.99	[5.05–6.93]
>3 years	68	120	37,790	3.18	[2.61–3.74]

* Significant differences in overall pain rate per 1000-training-hour exposure among the three age groups; 95% CI: 95% confidence interval.

## Data Availability

Not applicable.

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
