# Peer review of "The Characteristics of Badminton-Related Pain in Pre-Adolescent and Adolescent Badminton Players"

_children, 2023, doi:10.3390/children10091501_

Round 1
Reviewer 1 Report
Title: The characteristics of badminton-related pain in child badminton players
Main concerns:
- Title. “child” is not appropriate here. Please use the term “pre-adolescent and adolescent”. Also, this same term should be used throughout your manuscript.
- Line 127 – 128. What are the differences between “badminton experiences” and “training days and hours”? Aren’t they the same – training for badminton is exposure to badminton-play and thus is badminton experiences?
- Why does Fig 2 not include the ankle site since it was already established that ankle is the most reported region of pain? The discussion should then focus on why the ankle is the frequent site of pain relative to the shoulder, lower back and knee as was observe din other overhead sports. Authors should provide some possible reasons and also how to deal or mitigate with this issue – maybe strengthening or stretching of the lower limb joints specifically., how? Perhaps the ankle joint is not able to take the weight of the child who is jumping, pivoting, lunging during training?
Minor issues:
- English sentence structure and use of words throughout need to improve. Please get an English language expert to read your manuscript.
- Line 13-14. This sentence is awkward. Please re-write.
- Line 15. Delete “athlete” – because you are not working with athletes – use the term “training”. Same throughout the manuscript.
- - Line 32. Delete “one kind of racquet sports”.
- Line 42. Should be “Literature”.
- Line 82 & 85. “matching” is not correct here.
- Line 117. Write out the 60.9% in words.
- Line 194. "Pain" is spelled wrongly here.
- Line 203. Change “might” to “is”.
- Line 231. “…. we attempted to address potential sources of bias” How did you do this – it was not highlighted in the Methods section.
Nothing to add.
Author Response
Point to point responses to the reviewers’ comments
We are grateful for very helpful comments. We have revised the paper accordingly. The revised parts are underlined in the manuscript. The response to each comment is summarized in the following.
Reviewer 1
Main concerns:
Point number 1
Title. “child” is not appropriate here. Please use the term “pre-adolescent and adolescent”. Also, this same term should be used throughout your manuscript.
Our responses
We agree with the reviewer’ s comment and have rewritten the title. The term “pre-adolescent and adolescent” has been used throughout the manuscript as well.
Text change
1.The characteristics of badminton-related pain in pre-adolescent and adolescent badminton players
2.This study aimed to survey badminton-related pain distribution and prevalence in pre-adolescent and adolescent badminton players.
- These data might help target the high prevalence of pain in body regions in pre-adolescent and adolescent badminton players across ages for badminton injury prevention programs.
- This study aimed to survey the distribution and rate of badminton-related pain, especially knee pain, lower back pain, and shoulder pain, among pre-adolescent and adolescent badminton players participating in the national tournament level.
- A cross-sectional study was designed to investigate pre-adolescent and adolescent badminton players who ranged in age from 7 to 12 years. All the participants belonged to the Japan Schoolchildren Badminton Federation. We recruited 611 pre-adolescent and adolescent badminton players (260 boys and 351 girls) attending the national tournament games at three randomly selected locations in 2019.
6.The results showed that 60.9% of all the 366 pre-adolescent and adolescent badminton players aged 7-12 years experienced at least one pain associated with badminton, which is less than the pain rate (73%) in competitive and amateur badminton players over 18 years [19].
7.In respect to pain rate of badminton players, no studies using pain rate per 1000 badminton training-hours exposures have been found. Literature using injury rate of per 1000 athlete-hours exposures revealed that injury incidence rates increased with increasing age in pre-adolescent and adolescent athletes aged 7-12 years [28]. As well as injury rates, we found that pain rate of per 1000 training-hours exposures increased with increasing age: 1.22 in 7-8 year-old badminton players, 2.29 in 9-10 year-old badminton players, and 4.34 in 11-12 year-old badminton players. In addition, another interesting finding was found that, irrespective of age groups, pre-adolescent and adolescent badminton players with 2-3 years of badminton experience had the highest pain rate. Pre-adolescent and adolescent badminton players require almost one year of badminton skills learning and practice to develop themselves from stage to first supervised training, then another year to develop themselves from supervised training to first competition [29]. We speculated that competition participation will lead to subsequently high-intensity and aggressive training, even though pre-adolescent and adolescent badminton players present immature badminton skills, which will lead to the high prevalence of pain in the next year as a consequence. In future, badminton skill mastery and training loading among pre-adolescent and adolescent badminton players with 2-3 years of badminton experience should be studied.
8.In 7-12-year-old pre-adolescent and adolescent badminton players, badminton pain mostly involved the lower limbs. The prevalence of pain increased with age. These data might help target the high prevalence of pain in the body regions in pre-adolescent and adolescent badminton players across ages for badminton injury prevention programs.
Point number 2
Line 127 – 128. What are the differences between “badminton experiences” and “training days and hours”? Aren’t they the same – training for badminton is exposure to badminton-play and thus is badminton experiences?
Our response
“badminton experiences” is the years of badminton playing. We have written the sentence.
Text change
Among the three groups, significant differences in age (p < 0.001) and years of badminton playing experience (p < 0.001) were observed whereas no significant differences in training hours per day, training days per week, training hours per week, and training hours per year were observed.
Point number 3
- Why does Fig 2 not include the ankle site since it was already established that ankle is the most reported region of pain? The discussion should then focus on why the ankle is the frequent site of pain relative to the shoulder, lower back and knee as was observe din other overhead sports. Authors should provide some possible reasons and also how to deal or mitigate with this issue – maybe strengthening or stretching of the lower limb joints specifically., how? Perhaps the ankle joint is not able to take the weight of the child who is jumping, pivoting, lunging during training?
Our response
A majority of previous studies on overhead motion sports have demonstrated that there were associations between knee pain, lower back pain, and shoulder pain. Therefore, in this study, we mainly focused on the three anatomic sites. We agree with the reviewer’s comment, and have provide some possible reasons.
Text change
The following sentences were added:
Meanwhile, in badminton players, a significant association existed between shoulder pain, lower back pain, and knee pain [13], which means pain localized in one of the three anatomic sites will produce a load that has to be compensated for by movement of the other two sites. Nonetheless, studies on the epidemiology of pain related to badminton in knee, lower back, and shoulder among pre-adolescent and adolescent badminton players are finite. Additionally, studies on pain related to badminton using per 1000 badminton training-hours exposures have not been found [4].
Moreover, badminton players whose technique is immature present more internal joint rotation in the horizontal plane as well as more inversion joint moment in the frontal plane [26]. Plus, the participants of this study were pre-adolescent and adolescent badminton players whose physical fitness (such as muscle, bone, and neuro) was de-veloping, which means they might be unable to respond to intensive somatic demands of weight transfer repetitively through turning, pivoting, and landing, which produce extra load in the ankle joint during badminton training. In addition, vigorous training load causes fatigue which increases the risk of ankle sprain in badminton playing [27]. These findings could interpret why ankle was the most common pain site among pre-adolescent and adolescent badminton players.
With respect to static ability training, plyometric training on balance could improve the static balance ability [31], which might decrease the prevalence of ankle pain and knee pain.
26.Fu, L., Ren, F., Baker, J.S. Comparison of joint loading in badminton lunging between professional and amateur badminton players. Appl. Bionics. Biomech. 2017, 2017, 5397656.
27.Herbaut A, Delannoy J. Fatigue increases ankle sprain risk in badminton players: A biomechanical study. J. Sports Sci. 2020, 38, 1560-1565.
31.Karadenizli, Z.I. The Effects of Plyometric Education Trainings on Balance and Some Psychomotor Characteristics of School Handball Team. Univers. J. Educ. Res. 2016, 4, 2292-2299.
Minor issues:
Point number 4
English sentence structure and use of words throughout need to improve. Please get an English language expert to read your manuscript.
Our responses
The paper has been proofread by native check.
Point number 5
Line 13-14. This sentence is awkward. Please re-write.
Our responses
Thank you very much for the reviewer’s comment. We have rewritten the sentence.
Text change
Profiles of badminton-related pain were surveyed using a questionnaire among 366 pre-adolescent and adolescent badminton players aged 7-12 years. The distribution of badminton-related pain was described, and the pain incidence was calculated.
Point number 6
Line 15. Delete “athlete” – because you are not working with athletes – use the term “training”. Same throughout the manuscript.
Our responses
We agree with the reviewer’s comment and have changed the term throughout the manuscript.
Text change
1.Pain proportions and per 1000 training-hours exposures were the main outcome measures.
- Among all the participants, the pain rate per 1000 training-hours exposures was 3.06.
3.Additionally, studies on pain related to badminton using per 1000 badminton training-hours exposures have not been found.
- Poisson distribution was used to calculate the pain rate per 1000 training-hours exposures for comparing the pain incidences between the three groups. The 95% confidence interval (CI) of the pain rate per 1000 training-hours exposures was also calculated. An hour of badminton training exposures is defined as one hour of somatic condition training or badminton skills under the supervision of the coach without the time of warm-up and cool-down by one badminton player. We calculated the pain rate per 1000 training-hours exposures in the badminton training period as follows:
Pain rate per 1000 training-hours exposures = [∑(No. of pains)/∑{(No. of participants) × (hours of badminton training)}] × 1,000.
If the 95% CI did not overlap, significant differences in pain rate per 1000 training-hours exposures were assumed to exist between the groups. Statistically significant was defined as P < 0.05.
- The overall pain rate per 1000 training-hours exposures was 3.06 (95% CI: 2.81-3.32). All the pain incidences in face, chest, abdomen, head, neck, scapular, back, forearm, and hip were less than 2%, so the nine anatomic sites were gathered together in the category “others”. As shown in Figure 1, ankle was the most common pain site, 12.5% of all the pain, followed by knee (11.4%), plantar (10.1%), shoulder (8.1%), lower back (6.5%), toe (5.8%), groin (4.9%), quadriceps (4.9%), elbow (4.5%), calf (4.3%), Achilles tendon (4.3%), hamstring (4.0%), wrist (3.4%), shin (2.7%), and upper arm (2.2%). Further, to improve the understanding of pain related to badminton occur in knee, lower backer, and shoulder, the pain rates per 1000 training-hours exposures of the three anatomic sites were also calculated. The pain rates per 1000 training-hours exposures of shoulder pain, lower back pain, and knee pain are shown in Figure 2. The overall pain rate of the three sites was 0.80 (95% CI: 0.67-0.93) pain per 1000 training-hours exposures. The pain rates of shoulder pain, lower back pain, and knee pain were 0.25 (95% CI: 0.18-0.32) pain, 0.20 (95% CI: 0.13-0.26) pain, and 0.35 (95% CI: 0.26-0.43) pain per 1000 training-hours exposures.
7.Figure 2. Pain rate per 1000 training-hours exposure in shoulder, lower back, and knee.
8.Regardless of the three groups, participants with 2-3 years of badminton playing experience presented the highest pain rate per 1000 training-hours exposures.
- Table 2. Pain rates in different age groups broken down using badminton experience.
Variable |
Players |
Pain |
Training-hours of exposures |
Pain rate per 1000 training-hours exposures |
95% CI |
7-8 year-old |
|
|
|
|
|
Overall |
52 |
30 |
24670 |
1.22* |
[0.78-1.65] |
≤ 2 years |
37 |
17 |
16490 |
1.03 |
[0.54-1.52] |
2-3 years |
11 |
10 |
6220 |
1.61 |
[0.61-2.60] |
> 3 years |
4 |
3 |
1960 |
1.53 |
[0.00-3.26] |
9-10 year-old |
|
|
|
|
|
Overall |
155 |
173 |
75420 |
2.29* |
[1.95-2.64] |
≤ 2 years |
55 |
51 |
23990 |
2.13 |
[1.54-2.71] |
2-3 years |
49 |
68 |
23280 |
2.92 |
[2.23-3.62] |
> 3 years |
51 |
54 |
28150 |
1.92 |
[1.41-2.43] |
11-12 year-old |
|
|
|
|
|
Overall |
159 |
351 |
80920 |
4.34* |
[3.88-4.79] |
≤ 2 years |
37 |
75 |
17100 |
4.39 |
[3.39-5.38] |
2-3 years |
54 |
156 |
26030 |
5.99 |
[5.05-6.93] |
> 3 years |
68 |
120 |
37790 |
3.18 |
[2.61-3.74] |
*Significant differences in overall pain rate per 1000 training-hours exposures among the three age groups.
95% CI: 95% confidence interval.
- Moreover, this study is the first to study badminton-related pain in shoulder, lower back, and knee using pain rate of per 1000 training-hours in pre-adolescent and adolescent badminton players. The pain rate of per 1000 training-hours exposures was 0.25 in shoulder, 0.20 in lower back, and 0.35 in knee.
- In respect to pain rate of badminton players, no studies using pain rate per 1000 badminton training-hours exposures have been found. Literature using injury rate of per 1000 athlete-hours exposures revealed that injury incidence rates increased with increasing age in pre-adolescent and adolescent athletes aged 7-12 years [28]. As well as injury rates, we found that pain rate of per 1000 training-hours exposures increased with increasing age: 1.22 in 7-8 year-old badminton players, 2.29 in 9-10 year-old badminton players, and 4.34 in 11-12 year-old badminton players.
Point number 7
Line 32. Delete “one kind of racquet sports”.
Our responses
We agree with the reviewer’s comment and have deleted the sentence.
Text change
Badminton requires repetitive forehand overhead motion, lunges, jumps, quick directional changes, and all body joint movements in response to different types of strokes [3,4].
Point number 8
Line 42. Should be “Literature”.
Our responses
We agree with the reviewer’s comment and have rewritten the word.
Text change
Literature reported that among badminton players aged 6-18 years, 27.6% of them complained about shoulder pain and 35.4% of them complained about lower back pain; pain related to badminton was common in foot, knee, shoulder/elbow, and back [11,12].
Point number 9
Line 82 & 85. “matching” is not correct here.
Our response
Thank you very much for the reviewer’s comment. We have rewritten the word.
Text change
Pain was defined as any painful physical discomfort (ache or soreness in anatomical regions, without or with radiating pain) with sustained sports capability [20] as follows: (1) being able to continue the present badminton training or match; (2) participating in the next scheduled badminton training or match without time loss; (3) not need medical care during and after the badminton training or match. An injury was defined as any physical discomfort that caused one or more of the following three judgment criteria to be met during training or match play: (1) having to immediately stop the present badminton training or match; (2) being absent from the following badminton training or match; and/or (3) the need for medical attention regardless of the possibility of missing training or a match.
Point number 10
Line 117. Write out the 60.9% in words.
Our responses
We agree with the reviewer’s comment and have rewritten the word.
Text change
Sixty-nine percent (223 participants: 87 boys and 136 girls) of all the 366 participants experienced at least one pain related to badminton.
Point number 11
Line 194. "Pain" is spelled wrongly here.
Our responses
Thank you very much for the reviewer’s comment. We have rewritten the word.
Text change
On the contrary, previous studies of baseball reported that 35% of all the players experienced shoulder pain, which was more than 6.04% of pitching-related pain in 9-12 year-old American baseball players [11].
Point number 12
Line 203. Change “might” to “is”.
Our responses
We have changed the word.
Text change
Moreover, this study is the first to study badminton-related pain in shoulder, lower back, and knee using pain rate of per 1000 training-hours in pre-adolescent and adolescent badminton players.
Point number 13
Line 231. “…. we attempted to address potential sources of bias” How did you do this – it was not highlighted in the Methods section.
Our response
Thank you very much for the comment. The questionnaire design and the definition of injury and pain were considered for addressing the potential sources of bias. Also, the definition of training hours was designed for addressing the potential sources of bias.
1.All the pain and injuries were reported specifically in terms of 25 anatomical regions presenting in a body image.
- Pain was defined as any painful physical discomfort (ache or soreness in anatomical regions, without or with radiating pain) with sustained sports capability [20] as follows: (1) being able to continue the present badminton training or match; (2) participating in the next scheduled badminton training or match without time loss; (3) not need medical care during and after the badminton training or match. An injury was defined as any physical discomfort that caused one or more of the following three judgment criteria to be met during training or match play: (1) having to immediately stop the present badminton training or match; (2) being absent from the following badminton training or match; and/or (3) the need for medical attention regardless of the possibility of missing training or a match. A participant who reported injured or surgery sites without other pain sites was excluded. The injured or surgery sites were excluded once injured or surgery sites and other pain sites were both reported. Additionally, a participant with badminton experience of less than one year was excluded.
Text change
The following sentences were added:
We defined the time of badminton skill training or somatic training under the coach’s supervision as training hours. The duration of the warm-up and cool-down periods was not considered training exposure time.
Reviewer 2 Report
Line 51
Additionally, studies on pain related to badminton using per 1000 athlete-hours ex- 51 posures have not been found.
Coment 1
This is certainly the conclusion of a study, so it would be good to provide a reference.
Line 100
The normality of baseline parameters was examined by Shapiro-Wilk test.
Coment 2
Test results are not shown, it should at least be emphasized that the distribution is not normal and therefore non-parametric statistics are used.
Line 141-144
As shown in Figure 1, ankle was the most common pain site, 12.5% of all the pain, followed by knee (11.4%), plantar (10.1%), shoulder (8.1%), lower back (6.5%), toe (5.8%), groin (4.9%), quadriceps (4.9%), elbow (4.5%), calf (4.3%), Achilles tendon (4.3%), hamstring (4.0%), wrist (3.4%), shin (2.7%), and upper arm (2.2%).
Coment 3
These data are shown in the table below, there is no need to show them in text and table form.
Line 163
(7-8-year-old group: 1.61, 9-10-year-old group: 2.92, 11-12-year-old group: 5.99).
Coment 4
These data are also in the table.
Line 70-73
(including face, chest, abdomen, shoulder, elbow, wrist, finger, groin, quadriceps, knee, ankle, shin, toe, head, neck, scapular, back, upper arm, forearm, lower back, hip, hamstring, calf, Achilles tendon, and plantar)
There are 25 anatomical regions in the questionnaire
Figure 1 The distribution of pain in all the participants. 17 anatomical regions
Figure 2. Pain rate per 1000 badminton-hours exposure in shoulder, lower back, and 154 knee. The error bars represent 95% CI. 3 anatomical regions
Coment 5
There is a need to explain somewhere why the data is not shown for all regions.
Table 1 and Table 2
Coment 6
In Table 1 the statistically significant differences are marked with an asterisk, but in Table 2 they are not. It would be good to standardize this.
Table 1
Values are mean ± standard deviation. *p value < 0.001 among the three age groups.
Coment 7
There is a statistically significant difference in experience.
Author Response
Point to point responses to the reviewers’ comments
We are grateful for very helpful comments. We have revised the paper accordingly. The revised parts are underlined in the manuscript. The response to each comment is summarized in the following.
Reviewer2
Point number 1
Line 51
Additionally, studies on pain related to badminton using per 1000 athlete-hours exposures have not been found.
Coment 1
This is certainly the conclusion of a study, so it would be good to provide a reference.
Our response
We agree with the reviewer’s comment, and have added a reference.
Text change
Additionally, studies on pain related to badminton using per 1000 badminton training-hours exposures have not been found [4].
4.Pardiwala, D.N.; Subbiah, K.; Rao, N.; Modi, R. Badminton Injuries in Elite Athletes: A Review of Epidemiology and Bio-mechanics. Indian J. Orthop. 2020, 54, 237-245.
Point number 2
Line 100
The normality of baseline parameters was examined by Shapiro-Wilk test.
Coment 2
Test results are not shown, it should at least be emphasized that the distribution is not normal and therefore non-parametric statistics are used.
Our response
We agree with the reviewer’s comment, and have added some sentences.
Text change
The following sentences were added:
The data of the baseline parameters, including age, years of badminton playing experiences, badminton training duration of time per day, badminton training days weekly, badminton training hours weekly, and badminton training hours one year, presented an abnormal distribution.
Point number 3
Line 141-144
As shown in Figure 1, ankle was the most common pain site, 12.5% of all the pain, followed by knee (11.4%), plantar (10.1%), shoulder (8.1%), lower back (6.5%), toe (5.8%), groin (4.9%), quadriceps (4.9%), elbow (4.5%), calf (4.3%), Achilles tendon (4.3%), hamstring (4.0%), wrist (3.4%), shin (2.7%), and upper arm (2.2%).
Coment 3
These data are shown in the table below, there is no need to show them in text and table form.
Our response
Although Figure 1 shows the distribution of the anatomic regions, the readers are unable to get the pain incidence from the figure. Therefore, we think that the accurate data presented in the text are appropriate.
Text change
None.
Point number 4
Line 163
(7-8-year-old group: 1.61, 9-10-year-old group: 2.92, 11-12-year-old group: 5.99).
Coment 4
These data are also in the table.
Our response
Thank you very much for the comment. We have deleted the sentence.
Text change
Regardless of the three groups, participants with 2-3 years of badminton playing experience presented the highest pain rate per 1000 training-hours exposures.
Point number 5
Line 70-73
(including face, chest, abdomen, shoulder, elbow, wrist, finger, groin, quadriceps, knee, ankle, shin, toe, head, neck, scapular, back, upper arm, forearm, lower back, hip, hamstring, calf, Achilles tendon, and plantar)
There are 25 anatomical regions in the questionnaire
Figure 1 The distribution of pain in all the participants. 17 anatomical regions
Figure 2. Pain rate per 1000 badminton-hours exposure in shoulder, lower back, and knee. The error bars represent 95% CI. 3 anatomical regions
Coment 5
There is a need to explain somewhere why the data is not shown for all regions.
Our response
In this study, we investigated 25 anatomical regions. However, the pain incidences of 9 regions including face, chest, abdomen, head, neck, scapular, back, forearm, and hip were low, so we gathered them together in the category “others”. We agree with the reviewer’s comment and have explained.
Text change
The following sentences were added:
Meanwhile, in badminton players, a significant association existed between shoulder pain, lower back pain, and knee pain [13], which means pain localized in one of the three anatomic sites will produce a load that has to be compensated for by movement of the other two sites. Nonetheless, studies on the epidemiology of pain related to badminton in knee, lower back, and shoulder among pre-adolescent and adolescent badminton players are finite.
This study aimed to survey the distribution and rate of badminton-related pain, especially knee pain, lower back pain, and shoulder pain, among pre-adolescent and adolescent badminton players participating in the national tournament level.
All the pain incidences in face, chest, abdomen, head, neck, scapular, back, forearm, and hip were less than 2%, so the nine anatomic sites were gathered together in the category “others”.
Further, to improve the understanding of pain related to badminton occur in knee, lower back, and shoulder, the pain rates per 1000 training-hours exposures of the three anatomic sites were also calculated.
Point number 6
Table 1 and Table 2
Coment 6
In Table 1 the statistically significant differences are marked with an asterisk, but in Table 2 they are not. It would be good to standardize this.
Our response
We agree with the reviewer’s comment and have standardized Table 2.
Text change
Table 2. Pain rates in different age groups broken down using badminton experience.
Variable |
Players |
Pain |
Training-hours of exposures |
Pain rate per 1000 training-hours exposures |
95% CI |
7-8 year-old |
|
|
|
|
|
Overall |
52 |
30 |
24670 |
1.22* |
[0.78-1.65] |
≤ 2 years |
37 |
17 |
16490 |
1.03 |
[0.54-1.52] |
2-3 years |
11 |
10 |
6220 |
1.61 |
[0.61-2.60] |
> 3 years |
4 |
3 |
1960 |
1.53 |
[0.00-3.26] |
9-10 year-old |
|
|
|
|
|
Overall |
155 |
173 |
75420 |
2.29* |
[1.95-2.64] |
≤ 2 years |
55 |
51 |
23990 |
2.13 |
[1.54-2.71] |
2-3 years |
49 |
68 |
23280 |
2.92 |
[2.23-3.62] |
> 3 years |
51 |
54 |
28150 |
1.92 |
[1.41-2.43] |
11-12 year-old |
|
|
|
|
|
Overall |
159 |
351 |
80920 |
4.34* |
[3.88-4.79] |
≤ 2 years |
37 |
75 |
17100 |
4.39 |
[3.39-5.38] |
2-3 years |
54 |
156 |
26030 |
5.99 |
[5.05-6.93] |
> 3 years |
68 |
120 |
37790 |
3.18 |
[2.61-3.74] |
*Significant differences in overall pain rate per 1000 training-hours exposures among the three age groups.
95% CI: 95% confidence interval.
Point number 7
Table 1
Values are mean ± standard deviation. *p value < 0.001 among the three age groups.
Coment 7
There is a statistically significant difference in experience.
Our response
We have rewritten the sentence.
Text change
Among the three groups, significant differences in age (p < 0.001) and years of badminton playing experience (p < 0.001) were observed whereas no significant differences in training hours per day, training days per week, training hours per week, and training hours per year were observed.
Reviewer 3 Report
The study is merely descriptive. Despite that, it brings some body of knowledge to this sport. I strongly suggest to edit plots and figures.
Minor editing of English language required.
Round 2
Reviewer 3 Report
The badminton community will be happy if this work. Nice joob.